# Characterizing the ZrC(111)/*c*-ZrO_2_(111) Hetero-Ceramic Interface: First Principles DFT and Atomistic Thermodynamic Modeling

**DOI:** 10.3390/molecules27092954

**Published:** 2022-05-05

**Authors:** Eric Osei-Agyemang, Jean-François Paul, Romain Lucas, Sylvie Foucaud, Sylvain Cristol, Anne-Sophie Mamede, Nicolas Nuns, Ahmed Addad

**Affiliations:** 1Department of Materials Design and Innovation, University at Buffalo, Buffalo, NY 14260-1660, USA; ericosei@buffalo.edu; 2UMR 8181—UCCS—Unité de Catalyse et Chimie du Solide, CNRS, Centrale Lille, Univ. Artois, Université de Lille 1, F-59000 Lille, France; jean-francois.paul@univ-lille.fr (J.-F.P.); sylvain.cristol@univ-lille.fr (S.C.); anne-sophie.mamede@univ-lille.fr (A.-S.M.); nicolas.nuns@univ-lille.fr (N.N.); 3IRCER, UMR 7315, Université de Limoges, F-87068 Limoges, France; sylive.foucaud@unilim.fr; 4CNRS-UMR 8207, UMÉT, Unité MatÉriaux et Transformations, F-59000 Lille, France; ahmed.addad@univ-lille.fr

**Keywords:** ZrC, DFT, thermodynamic

## Abstract

The mechanical and physical properties of zirconium carbide (ZrC) are limited to its ability to deteriorate in oxidizing environments. Low refractory oxides are typically formed as layers on ZrC surfaces when exposed to the slightest concentrations of oxygen. However, this carbide has a wide range of applications in nuclear reactor lines and nozzle flaps in the aerospace industry, just to name a few. To develop mechanically strong and oxygen-resistant ZrC materials, the need for studying and characterizing the oxidized layers, with emphasis on the interfacial structure between ZrC and the oxidized phases, cannot be understated. In this paper, the ZrC(111)//*c*-ZrO_2_ (111) interface was studied by both finite temperature molecular dynamic simulation and DFT. The interfacial mechanical properties were characterized by the work of adhesion which revealed a Zr|OO|Zr|OO//ZrC(111) interface model as the most stable with an oxygen layer from ZrO_2_ being deposited on the ZrC(111) surface. Further structural analysis at the interface showed a crack in the first ZrO_2_ layer at the interfacial region. Investigations of the electronic structure using the density of state calculations and Bader charge analysis revealed the interfacial properties as local effects with no significant impacts in the bulk regions of the interface slab.

## 1. Introduction

Zirconium Carbide (ZrC), being a non-oxide ultra-high temperature ceramic, is an interesting material for several applications in severe environments. It is mostly used in environmentally harsh and demanding conditions including cutting tools, nuclear plant inner coatings, turbine components in the aerospace industry, and as refractory ceramics in the steel industry [1,2]. It fulfills all these application requirements due to its excellent mechanical and physical properties with a high melting point of 3430 °C.

However, a serious problem with the oxidation of ZrC was encountered with this material when used in harsh conditions. Indeed, ZrC forms low refractory oxides at 500–600 °C [3]. The oxide causes deterioration of the physical and mechanical properties and defeats the purpose of its general applications. Thus, there is a need to study and properly understand the oxidation process and mechanism on the low index surfaces of this nanocrystallite material. ZrC, being cubic, has three distinct low index surfaces: (100), (110), and (111) surfaces, with (100) being the most stable [4,5]. Studies on the oxidation process have been carried out on the (100) surface [6,7,8,9] and all provide similar results: the ZrC(100) surface is extremely reactive to oxygen and easily oxidized by the smallest concentration of oxygen. In a recent study, we observed that the ZrC(110) surface was easily oxidized with the formation of ZrO_2_ on the exposed surface [10]. In another experimental study, the (110) orientation of *t*-ZrO_2_ was described, and the results showed that it grows preferentially on the ZrC(100) surface [11,12]. Similar experiments [13,14,15,16] and theoretical studies [17] on the oxidation of the ZrC(111) surface have been conducted by several research groups. In all these investigations, oxygen was observed to be very reactive on the (111) surface and dissociated completely into atomic species.

Cleaving ZrC bulk along the (111) normal plane produces a crystal with Zr and C layers terminating on opposite sides of the exposed surfaces. This renders the slab polar. However, experimental findings show the surface to terminate preferentially with Zr layers [18,19,20]. In a previous theoretical study, several reconstructions were carried out on the ZrC(111) surface with different terminations. The most thermodynamically stable surface was terminated with four Zr atoms [4]. Thus, this surface slab contained excess Zr atoms over C atoms.

In parallel, theoretical studies have been carried out to study the heteroceramic interface of ZrO_2_ deposited on α-Al_2_O_3_($1\bar{1}02)$ substrates and found the stoichiometric ZrO_2_(001)/α-Al_2_O_3_($1\bar{1}02)$ interface to be weakly bonded, regardless of the film thickness [21]. However, in another work where the ZrO_2_(111) ceramic was deposited on a metal (Ni) substrate, it was observed to adhere very strongly at the monolayer level, but thicker films of the ceramic interacted very weakly with the Ni substrate [22]. ZrO_2_ was overly observed to transform partially into *m*-ZrO_2_ but could not completely convert into *m*-ZrO_2_ due to the constraints imposed by the periodic boundary conditions.

To reduce the susceptibility of ZrC to easy oxidation, one method can be to coat the surface with another ceramic material (e.g., SiC) through the synthesis of hybrid objects, which leads to a protective layer on ZrC in oxidizing environments [23,24,25]. As the composition of the surface will condition the further functionalization that could involve preceramic precursors, such as polycarbosilanes, it is necessary to study the oxidized layer. This will provide a detailed analysis of the structure, energy, and stability of the interfacial region between ZrC and the oxidized layer. Thus, to answer this need of understanding the interface of ZrC-ZrO_2_, this paper is organized as follows: Section 2 provides details on the finite temperature molecular dynamic simulation and general calculation parameters as well as procedures for building the interface. Section 3 deals with the results obtained and discusses them accordingly. Finally, Section 4 gives the summary and conclusions of the current study.

## 2. Structural Models and Calculation Schemes

### 2.1. General Computational Details

Since ZrC was used as the substrate, all calculation parameters were based on the optimized values for ZrC. Density functional theory (DFT) was performed using the Vienna ab initio Simulation Package (VASP) [26] based on Mermin’s finite temperature DFT [27]. The following electronic configurations were used for Zr, C, and O atoms, respectively: [Kr]4d^2^5s^2^, [He]2s^2^2p^2^, and [He]2s^2^2p^4^. We used Projected Augmented Wavefunction (PAW) [28] to describe the core electrons and the core part of the valence electrons wavefunctions and this helped in reducing the number of plane waves required to describe the electrons close to the nuclei. The Kohn-Sham valence states were expanded in a plane wave basis set with a kinetic energy cutoff of 500 eV. The generalized gradient approximation (GGA) parametrized by Perdew, Burke, and Ernzerhof (PBE) [29] was used for the exchange correlation part. Dispersion corrections have not been added to the DFT energies as, at the interfaces, covalent bonds are formed and the atomic density of the two solids are comparable. The interface interaction may be slightly overestimated but the stability order and the conclusions will not be influenced. The Methfessel-Paxton [30] smearing scheme was utilized with the gamma parameter set to 0.1 eV. For all bulk calculations, a k-sampling of a 9 × 9 × 9 mesh using the standard Monkhorst-Pack [31] special grid was employed. However, 9 × 9 × 1 k-points sampling was selected for all surface and interface slab calculations. The Kohn-Sham equations were resolved using the self-consistent field (SCF) procedure and assumed to be converged when energy changes of 1 × 10^−4^ eV were obtained between two successive iterations.

### 2.2. Finite Temperature Molecular Dynamics (MD)

To confirm the experimental findings on the analysis of ZrC nanocrystals, finite temperature ab initio molecular dynamic simulation was performed to provide a first approximation on the nature of ZrO_2_ formed on the ZrC surfaces. A (2 × 2) supercell was used for all MD simulations (13.40 × 13.40 × 30.00 Å, 60 atoms). It started with a nine-layer thick ZrC(111) substrate (terminating with four Zr atoms on both sides of the surface slab) by depositing Zr and O atoms onto the exposed ZrC(111) surface to form about two layers of ZrO_2_. The ions were initially kept at T = 100 K within the micro canonical ensemble (NVE) and the velocities were scaled upwards at different steps until a final temperature of 1000 K was reached. This temperature was selected to provide an allowance for the possible formation of the *m*-ZrO_2_ phase which is stable at temperatures below 1450 K. A 1 fs time step was used. The resulting equilibrium structure after 20 ps was then quenched from 1000 K to 500 K. Geometries at minima on the potential energy surface were selected and optimized at a higher precision of calculation to obtain a final structure.

### 2.3. Bulk ZrC and *c*-ZrO_2_ Phases

To obtain parameters optimized for the system, bulk calculations were performed on ZrC and *c*-ZrO_2_ phases.

Energy versus volume data were obtained for both ZrC and *c*-ZrO_2_ and finally fitted with Murnaghan’s equation of state. The optimized lattice parameters were calculated from this fitting.

An optimized k-points of 5 × 5 × 5 Monkhorst-Pack grid producing 63 irreducible k-points was used for bulk ZrO_2_ calculations and the same kinetic energy cut-off of 500 eV for the ZrC bulk was used for the ZrO_2_ bulk. All the *c*-ZrO_2_ bulk used contained four formula units as shown in Figure 1.

### 2.4. Interface Model Construction

To construct the interface model, the stacking direction at the interface is initially selected and there should be a proper commensurability factor between the two bulk phases with respect to the interface plane [32]. The required surfaces are subsequently cleaved from the two bulk phases along the selected surface normal. Each of the revealed surfaces will have different atomic arrangements and configurations. The interface is finally created by bringing the two surfaces in contact with each other and then fully relaxed to obtain a final optimized interfacial geometry.

#### 2.4.1. Commensurate Phases and Surface Structures

Among the low index ZrC surfaces, the (100) stoichiometric and non-polar surface is found to be the most stable [4,5]. However, even though the (111) surface is polar upon cleaving from the bulk phase by terminating on one side with a carbon layer and the other side with a zirconium layer, surface reconstruction revealed a more stable surface terminated on both sides with Zr atom layers [4]. With a lattice parameter of 6.698 Å, *b* of 5.801 Å, and *β* = 60°, the exposed surface area of the ZrC(111) surface is 38.854 Å^2^. On accounts of several studies made on the *c*-ZrO_2_ surfaces, the (111) surface is found to be the most stable [33,34]. Surface energies are calculated for one layer up to six layers of ZrC to ascertain the effect of the layer thickness on the surface energy. The surface energies are calculated as $E_{surf}=(1/2A)\;\left[E_{slab}-nE_{bulk}\right]$ where *E_slab_* is the total energy of the surface slab, Ebulk is the energy per formula unit of ZrC in the corresponding bulk, *A* is the surface area, and *n* is the number of formula units in the surface slab.

Surface energies are also computed for the (111) terminations of *c*-ZrO_2_. The surface energies were calculated for different numbers of layers, starting from one to six layers of ZrO_2_. Upon cleaving the *c*-ZrO_2_ along the [111] direction, a polar slab is obtained with an OO layer terminating on one side and a Zr layer terminating on the other side. The slab can, however, be terminated in three different arrangements as OO|Zr|OO|Zr|OO-, Zr|OO|Zr|OO|Zr–, and O|Zr|OO|Zr|O– (Figure 1). Only symmetric slabs were used (slabs with mirror symmetry) in the calculation of the surface energy to eliminate the net dipole moment. The calculation of the interface tension defined in a subsequent section requires these surface energies. Thus, surface energies of three different terminations were calculated: Zr-termination, O-termination, and OO-termination.

To access surface energies of both stoichiometric and non-stoichiometric slabs, the surface grand potential is defined as *Ω^i^* which implies contact of the Zr and O reservoirs with the surface:(1)$$\Omega^{i}=\frac{1}{2A}\;\left[E_{slab}^{i}-N_{Zr}\mu_{zr}-N_{O}\mu_{O}\right]$$

*N_Zr_* and *N_O_* are the numbers of Zr and O atoms in the slab with *µ_Zr_* and *µ_O_* being the chemical potential of Zr and O, respectively. $E_{slab}^{i}$ is the total energy of the surface slab and *A* is the surface area. The chemical potentials of Zr and O are related by bulk ZrO_2_ in the expression: $\mu_{ZrO_{2}}=E_{ZrO_{2}}^{bulk}=\mu_{Zr}+2\mu_{O}$ with $E_{ZrO_{2}}^{bulk}$ being the total energy per bulk ZrO_2_ unit. Rearranging this expression and substituting it in Equation (1), we obtain the following:(2)$$\Omega^{i}=\frac{1}{2A}\;\left[E_{slab}^{i}-N_{Zr}E_{ZrO_{2}}^{bulk}-N_{O}\mu_{O}+2N_{Zr}\mu_{O}\right]$$

Defining the chemical potential of O in relation to the chemical potential of the reference state, O_2_ which is defined as half the total energy of O_2_ gas as ∆*µ_O_* = *µ_O_* − $\left(\frac{E_{O_{2}}^{gas}}{2}\right),$ and substituting it in Equation (2) with further rearrangements, we obtain:(3)$$\Omega^{i}=\frac{1}{2A}\;\left[E_{slab}^{i}-N_{Zr}E_{ZrO_{2}}^{bulk}+E_{O_{2}}^{gas}\left(N_{Zr}-\frac{N_{O}}{2}\right)+∆\mu_{O}\left(2N_{Zr}-N_{O}\right)\right]$$

If we make the following definition:(4)$$\gamma^{i}=\frac{1}{2A}\;\left[E_{slab}^{i}-N_{Zr}E_{ZrO_{2}}^{bulk}+E_{O_{2}}^{gas}\left(N_{Zr}-\frac{N_{O}}{2}\right)\right]$$
where *γ^i^* is the surface energy of the stoichiometric part of the selected slab. Substituting Equation (4) into Equation (3), we come to the following expression for the surface grand potential:(5)$$\Omega^{i}=\gamma^{i}+\frac{1}{2A}\;\left[∆\mu_{O}\left(2N_{Zr}-N_{O}\right)\right]$$

Thus, the surface grand potential is defined in terms of the surface energy arising from the stoichiometric part of the slab, and another part correcting for the extra number of Zr or O atoms.

From Equation (5), a range of $∆\mu_{O}$ values can be accessed if we define the lower and upper limits. In defining the upper limit of the O chemical potential, we assume that the chemical potential of O must be lower than the energy of O in its reference stable gaseous state. Thus comes the upper limit of the O chemical potential:(6)$$∆\mu_{O}=\mu_{O}-\frac{E_{O_{2}}^{gas}}{2}<0$$

For the lower limit of the O chemical potential, if we combine the expressions $\mu_{ZrO_{2}}=E_{ZrO_{2}}^{bulk}=\mu_{Zr}+2\mu_{O}$ with $∆\mu_{Zr}=\mu_{Zr}-E_{Zr}^{bulk}$ and $∆\mu_{O}=\mu_{O}-\left(E_{O_{2}}^{gas}/2\right)$, and make rearrangements, the lower limit of the O chemical potential is obtained:(7)$$∆\mu_{O}>\frac{1}{2}\;E_{ZrO_{2}}^{f}$$

$E_{ZrO_{2}}^{f}$ is the formation energy of ZrO_2_ defined as $E_{ZrO_{2}}^{f}=E_{ZrO_{2}}^{bulk}-E_{Zr}^{bulk}-E_{O_{2}}^{gas}$ and we calculated it as −9.97 eV. Thus, the range of potential accessible chemical values of O is:(8)$$-4.98\;\mathrm{eV}<∆\mu_{O}<0\ldots \mathrm{eV}$$

A plot of the surface grand potential *Ω^i^* against the accessible range of O chemical potential is obtained for both stoichiometric and non-stoichiometric slabs for easy comparison of surface energies.

Phase commensurability is one major problem that is encountered when forming interfaces. The two surfaces used in forming the interface must be coherent due to the periodic boundary condition imposed in the calculation. The surface misfit parameter ϒ can be used to obtain highly coherent interfaces [35]. This parameter is defined as:(9)$$ϒ=1-\frac{2S_{A-B}}{S_{A}+S_{B}}$$

Thus, a unit cell of *c*-ZrO_2_ with a surface area of *S_B_* is forced into coherency onto a substrate ZrC(111) with surface area *S_A_*, and the resulting overlap area between the two surfaces is *S_A–B_*. The misfit parameter measures the average length scale misfit between the two-unit cells [21] rather than an area misfit. In Table 1, the calculated misfit parameters between ZrC(111) substrate surface and all *c*-ZrO_2_ surfaces are summarized. It is apparent from this table that the ZrC(111)||*c*-ZrO_2_(111) interface combination has the lowest misfit parameter of 8.2% and is acceptable. The resulting interface unit cell defined by the substrate ZrC(111) is 6.698 Å × 6.698 Å which is small and can be easily managed by the DFT calculation.

The misfit parameter, being a geometrical measure, cannot be used alone in building the interface. It must be combined with other models. Two models are widely known for ensuring the commensurability of two different phases when forming an interface. Within the first approach, the unit cells of the two phases are multiplied by a factor corresponding to the other unit cell until both cells are commensurate with each other. The resulting supercell is usually large and unbearable for ab initio calculations. However, the resulting interface is coherent with very small mismatch parameters [36].

The second method is widely used [37,38,39,40,41,42] as it results in small and manageable interface supercells (a single unit cell), suitable for ab initio calculations. In this model, the lattice parameters of the phase considered as the substrate are considered for the interface, with the lattice parameter of the other phase scaled until a perfect match with the substrate lattice is obtained.

#### 2.4.2. Geometrical Models for Interface

Within the slab model described for studying the interface, a thickness of 10.945 Å of ZrC (nine layers) was used. This thickness was enough to mimic the electronic structure when ionic positions in the bulk are relaxed. The *c*-ZrO_2_(111) units were then pinned into the registry, layer by layer on the exposed ZrC(111) surface. Thus, in straining the *c*-ZrO_2_ to match the dimensions of the ZrC surface, coherent interfaces are ensured. The interface unit cell is therefore determined by the bulk and surface parameters of the ZrC(111). In this manner, the unit cell lattice parameter of the *c*-ZrO_2_(111) is shrunk by about 8%. After fixing the geometries of the two surfaces at the interface, the remaining degrees of freedom in the resulting interface structure are perpendicular to the interface and the interface chemical composition [43]. From one to five layers of the *c*-ZrO_2_(111) units were built on the ZrC(111) surface.

In Figure 2, we provide side views of the interface models used with the different numbers of ZrO_2_ layers. Each *c*-ZrO_2_ bilayer is approximately 3.5 Å thick. All interface models used were symmetric with respect to the center of the interface slab to remove any long-range dipole-dipole interaction between exposed surfaces. A total of 14 Å of vacuum was applied between two subsequent interface slabs to avoid any physical interactions between the slabs. The interface slab used thus has a configuration of: —ZrC(111)|*c*-ZrO_2_(111)|vacuum|ZrC(111)|*c*-ZrO_2_(111)|vacuum|ZrC(111)|*c*-ZrO_2_(111)|vacuum|ZrC(111)— Since the ZrC(111) slab is a reconstructed structure with four extra Zr atoms, the interface chemical composition is dependent on the number of Zr atoms (ZrC side) and the terminating layer of the *c*-ZrO_2_(111) phase. In constructing the interface, three different terminations along the *c*-ZrO_2_[111] direction were considered: Zr|OO|Zr|OO|Zr|OO—, O|Zr|OO|Zr|OO|Zr|O—, and OO|Zr|OO|Zr|OO|Zr—. We also considered the OO|Zr|OO|Zr|OO|Zr— on a ZrC(111) surface with an oxidized layer. A total of four different interface models were built as shown in Figure 2.

#### 2.4.3. Mechanics and Cohesion at the Interface

In defining the interface cohesion and stability, one important parameter used is the interface tension ɣ*_int_*, defined as the reversible work needed to separate the interface into two free surfaces [44]. According to this definition, an assumption made is that both diffusional and plastic degrees of freedom are suppressed and hence negligible. The greater the ɣ*_int_* value, the higher the energy needed to separate the interface into two surfaces.

The interface tension can be defined according to the Dupre equation in terms of the interface and free surface energies as [45,46]:(10)$${ɣ}_{int}=\sigma_{ZrC}+\sigma_{c-ZrO_{2}}-\sigma_{ZrC||c-ZrO_{2}}$$

$\sigma_{ZrC||c-ZrO_{2}}$ is the interface energy also known as the adiabatic work of adhesion, *W_ad_* > 0, $\sigma_{ZrC}$, and $\sigma_{c-ZrO_{2}}$ are the relaxed surface energies of the ZrC(111) and *c*-ZrO_2_(111) surfaces, respectively. In this definition, the relative strength of the interface versus the bulk bonds decides the preference for the formation of either the interface or the open surfaces [22].

A measure of whether there is the formation of an interface or the free surfaces can be determined by the interface tension. The magnitude and sign of ɣ*_int_* (Equation (10)) provide a measure of whether the interface bonds are stronger than the internal bonds in the separate phases [22]. The criteria are that 0 < ɣ*_int_* < $\sigma_{ZrC}+\sigma_{c-ZrO_{2}}$ corresponds to weakly coupled interfaces and ɣ*_int_* < 0 to strongly coupled interfaces. The calculated values of $\sigma_{ZrC}\;\mathrm{and}\;\sigma_{c-ZrO_{2}}$ used here are obtained from their respective relaxed bulk equilibrium phases (i.e., strain-free surface slabs).

In parallel, the adiabatic work of adhesion *W_ad_* is defined as:(11)$$W_{ad}=\frac{E_{ZrC}^{tot}+E_{c-ZrO_{2}}^{tot}-E_{ZrC||c-ZrO_{2}}^{tot}}{2A}$$

$E_{ZrC||c-ZrO_{2}}^{tot}$ is the total energy of the fully relaxed interface slab, *A* is the interface area, and $E_{ZrC}^{tot}\;\mathrm{and}\;E_{c-ZrO_{2}}^{tot}$ are the total energies of the fully relaxed isolated ZrC(111) and *c*-ZrO_2_(111) slabs, respectively. Usually, the calculated Wad value is a lower bound as compared to values obtained in cleavage experiments due to dissipative processes in physically separating the interface [44] There is no relation between characterizing the interfacial strength and the bulk strain when depositing the *c*-ZrO_2_. Hence, the $E_{c-ZrO_{2}}^{tot}$ value used is the total energy of the strained *c*-ZrO_2_ for commensurability with the ZrC surface. In this manner, the strain energy component between $E_{c-ZrO_{2}}^{tot}$ and $E_{ZrC||c-ZrO_{2}}^{tot}$ is canceled out since the *c*-ZrO_2_ is in the strain state [22].

Aside from the relaxed work of adhesion, the rigid work of adhesion $W_{ad}^{rigid}$ can be used in characterizing the interface cohesion and stability. In this definition, the same strained state is ensured to exist in both the interface and the free surfaces. This provides maximum cancelation for the strain energy in the calculated interface energy [43]. This quantity gives information purely on the bonds formed at the interface irrespective of the free surfaces. It is calculated by separating the optimized interface structure into the different phases and rigidly calculating their energies without allowing the phases to fully relax. Equation (11) is finally applied in calculating the rigid work of adhesion.

### 2.5. Interfacial Thermodynamics

For each couple of interface models used, the most stable chemical composition of the interface is determined by a thermodynamic approach. A thermodynamic grand canonical ensemble treatment is used to compare the relative stability of the models with different chemical compositions. Within such an ensemble, all models are assumed to be in chemical and thermal equilibria with bulk phases and the relevant thermodynamic quantity is the grand potential. An assumption made is that the entropic and volumetric contributions to the grand potential are negligible. For an interface of ZrC and *c*-ZrO_2_, the interface grand potential is defined as:(12)$$\Omega_{int}^{i/j}=\frac{1}{2}\left[\Omega_{slab}^{i/j}-N_{ZrC}\Omega_{ZrC}-N_{ZrO_{2}}\Omega_{ZrO_{2}}\right]-\Omega_{surf}^{ZrO_{2}}$$
where $\Omega_{int}^{i/j}$ is the interface grand potential, $\Omega_{surf}^{ZrO_{2}}$ is the surface grand potential of the exposed *c*-ZrO_2_ side of the interface slab and $\Omega_{slab}^{i/j}$, $\Omega_{ZrC}$, and $\Omega_{ZrO_{2}}$ are the grand potential of the interface slab, ZrC slab, and ZrO_2_ slabs, respectively. *N_ZrC_* and $N_{ZrO_{2}}$ are the number of ZrC and ZrO_2_ units in the respective slabs. Substituting the following:$$\begin{array}{c} \Omega_{slab}^{i/j}=E_{slab}^{i/j}-\sum_{k}\mu_{k}N_{k},\;\Omega_{ZrC}=E_{bulk}^{ZrC}-\mu_{ZrC},\;\Omega_{ZrO_{2}}=E_{bulk}^{ZrO_{2}}-\mu_{ZrO_{2}},\;\\ \mu_{ZrC}=\mu_{Zr}+\mu_{C}\;\mathrm{and}\;\mu_{ZrO_{2}}=\mu_{Zr}+2\mu_{O} \end{array}$$
into Equation (12), we obtain:(13)$$\begin{array}{c} \Omega_{int}^{i/j}=\frac{1}{2}[E_{slab}^{i/j}-N_{ZrC}E_{ZrC}^{bulk}-N_{ZrO_{2}}E_{ZrO_{2}}^{bulk}-\mu_{Zr}\left(N_{Zr}-N_{ZrC}-N_{ZrO_{2}}\right)\\ -\mu_{O}\left(N_{O}-2N_{ZrO_{2}}\right)-\left(E_{ZrC}^{bulk}-\mu_{Zr}\right)\left(N_{C}-N_{ZrC}\right)]-\Omega_{surf}^{ZrO_{2}} \end{array}$$
where $\mu_{Zr},\;\mu_{C},\;\mathrm{and}\;\mu_{O}$ are the chemical potentials of Zr, C, and O, respectively. $N_{k}$ is the number of that species, $E_{slab}^{i/j}$ is the total energy of the interface slab, $E_{bulk}^{ZrC}$ and $E_{bulk}^{ZrO_{2}}$ are the bulk energies of ZrC and ZrO_2_, respectively. Upon rearrangement of Equation (13), we obtain the following:(14)$$\begin{array}{c} \Omega_{int}^{i/j}=\frac{1}{2}[E_{slab}^{i/j}-N_{C}E_{ZrC}^{bulk}-N_{ZrO_{2}}E_{ZrO_{2}}^{bulk}-\mu_{Zr}\left(N_{Zr}-N_{C}-N_{ZrO_{2}}\right)\\ -\mu_{O}\left(N_{O}-2N_{ZrO_{2}}\right)]-\Omega_{surf}^{ZrO_{2}} \end{array}$$

If we make the following definition, $∆\mu_{O}=\mu_{O}-\mu_{O}^{*}$, and $∆\mu_{Zr}=\mu_{Zr}-\mu_{Zr}^{*}$ with $\mu_{Zr}^{*}=E_{Zr}^{bulk}\;\mathrm{and}\;\mu_{O}^{*}=E_{O_{2}}^{gas}/2$ then substituting in Equation (14) with subsequent rearrangement, we obtain:(15)$$\begin{array}{cc} \Omega_{int}^{i/j}=\frac{1}{2}[E_{slab}^{i/j} & -N_{ZrO_{2}}E_{ZrO_{2}}^{bulk}-N_{C}E_{ZrC}^{bulk}-E_{Zr}^{bulk}\left(N_{Zr}-N_{C}-N_{ZrO_{2}}\right)\\ & -\frac{1}{2}E_{O_{2}}^{gas}\left(N_{O}-2N_{ZrO_{2}}\right)-∆\mu_{Zr}\left(N_{Zr}-N_{C}-N_{ZrO_{2}}\right)-∆\mu_{O}(N_{O}\\ & -2N_{ZrO_{2}})]-\Omega_{surf}^{ZrO_{2}} \end{array}$$

If we define another quantity:(16)$$\begin{array}{cc} \emptyset_{int}^{i/j}=\frac{1}{2A}[E_{slab}^{i/j} & -N_{ZrO_{2}}E_{ZrO_{2}}^{bulk}-N_{C}E_{ZrC}^{bulk}-E_{Zr}^{bulk}\left(N_{Zr}-N_{C}-N_{ZrO_{2}}\right)\\ & -\frac{1}{2}E_{O_{2}}^{gas}\left(N_{O}-2N_{ZrO_{2}}\right)]-(\Omega_{surf}^{ZrO_{2}}/A) \end{array}$$

Substituting Equation (16) into Equation (15), we obtain an expression for the interface grand potential as:(17)$$\gamma_{int}^{i/j}=\frac{1}{A}\Omega_{int}^{i/j}=\emptyset_{int}^{i/j}+\frac{1}{2A}\left[∆\mu_{Zr}\left(N_{C}+N_{ZrO_{2}}-N_{Zr}\right)+∆\mu_{O}\left(2N_{ZrO_{2}}-N_{O}\right)\right]$$

Thus, for each overlayer termination i, the interface grand potential $\Omega_{int}^{i/j}$ depends on $∆\mu_{O}$ and $∆\mu_{Zr}$. A derivation of the upper and lower boundaries of the O and Zr chemical potentials is provided here as:
-For $∆\mu_{Zr}$, the upper boundary is $∆\mu_{Zr}=\mu_{Zr}-E_{Zr}^{bulk}<0$ and lower boundary is ∆μZr>12(EZrCf+EZrO2f) where $E_{ZrC}^{f}\;\mathrm{and}\;E_{ZrO_{2}}^{f}$ are the formation energies of ZrC and *c*-ZrO_2_, respectively. Thus, the Zr chemical potential range is defined as:
(18)$$-5.78\;\mathrm{eV}<∆\mu_{Zr}<0\;\mathrm{eV}$$-For O chemical potential, the upper limit is ∆μO=μO−12(EO2gas)<0, the lower boundary is $∆\mu_{O}>\left(E_{ZrO_{2}}^{f}/2\right),$ and the range of chemical potentials for O is:
(19)$$-4.98\;\mathrm{eV}<∆\mu_{O}<0\;\mathrm{eV}$$

## 3. Results and Discussion

### 3.1. ZrC-ZrO_2_: From Experiments to Theoretical Studies

As already described in previous papers [23,24,25] and according to TEM-ED experiments, the presence of a different phase from ZrC bulk was noticed at the surface of the particles. An EDX analysis revealed this phase to be zirconium oxide with an estimated thickness of 5 nm. Two different orientations were observed according to high image resolution (polycrystalline oxide layer). The dhkl indexation of the diffraction patterns of the different orientations showed the presence of cubic ZrO_2_, mainly the (111) phase with some traces of the tetragonal ZrO_2_ (101 and 011) phases (see Appendix A). To more precisely investigate the interface, a theoretical point of view may therefore be necessary.

### 3.2. Finite Temperature Molecular Dynamic Simulation

This section discusses the results obtained during the MD simulation. A haphazard ZrO_2_ structure was observed to grow on the ZrC(111) surface at 1000 K. This structure shows O atoms forming three-fold hollow bonds between three Zr atoms of the ZrC(111) surface at the interface. Upon quenching to a T = 500 K, a more ordered structure was obtained (Figure 3). The observed pattern of the ZrO_2_ atomic arrangements matches the crystal structure of *c*-ZrO_2_(111) and *t*-ZrO_2_(101) structures. Thus, the MD simulation confirms the formation of ZrO_2_ on the ZrC(111) surface from atomic depositions. This further complements the experimental results.

### 3.3. Surface and Bulk Properties of ZrC and ZrO_2_

Details of the optimized lattice parameter, bulk modulus as well as the pressure derivative of the bulk modulus of ZrC, are provided in a previous paper [47]. The optimized lattice parameter for the ZrC bulk is 4.736 Å. The fitted lattice parameter for *c*-ZrO_2_ is 5.143 Å. For *t*-ZrO_2_, the calculated a parameter is 3.649 Å and c = 5.257 Å and the tetragonal distortion dz = ∆z/c = 0.050. For *m*-ZrO_2_, the calculated values are a = 5.243 Å, b = 5.307 Å, c = 5.412 Å, and β = 99.20o. All these bulk parameters are well reproduced and are in excellent agreement with both experimental and other calculated values.

In a previous paper [4], we calculated the surface energy for a ZrC(111) surface terminating with four Zr atoms on both sides of the exposed surface at the different chemical potentials of C. The surface energy was 0.169 eV Å^−2^ at $\mu_{C}=E_{C}^{bulk}$. In Figure 4, a stability plot is provided for the surface grand potential of each of the surface terminations of *c*-ZrO_2_(111). The surface termination O|Zr|OO|Zr|O- is observed to be the most stable. The same stable termination is found in a different theoretical work [31].

For six layers of *c*-ZrO_2_(111) with the O|Zr|OO|Zr|O–termination, the calculated surface grand potential at $\mu_{O}=E_{O_{2}}^{Gas}/2$ is 0.054 eV Å^−2^ which agrees very well with 0.048 eV Å^−2^ in different works [35]. This is not surprising as this surface termination is the only stoichiometric structure used. Focusing on the most stable surface termination, surface energies were calculated with different numbers of ZrO_2_ layers. Table 2 provides a summary of the surface energies with different numbers of layers. The surface energies of the *c*-ZrO_2_(111) surface with O|Zr|OO|Zr|O– termination converges after three layers. Thus, the calculated surface energy of 0.054 eV Å^−2^ was used to compute the interface tension in Equation (10).

### 3.4. Interface Cohesion and Structure

#### 3.4.1. Rigid Work of Adhesion

Analysis of the interfacial cohesion begins with the rigid work of adhesion, previously defined in Section 2.4.3. With the rigid work of adhesion, the bulk properties of ZrC and *c*-ZrO_2_ are canceled out and the resulting parameter depends solely on interfacial properties. Table 3 provides a summary of the rigid work of adhesion values. In this table, it is apparent that the rigid work of adhesion is always lower than the relaxed work of adhesion for nearly all the interface models considered. Thus, the relaxation of the interface structure in the relaxed work of adhesion contributes significantly to the interface properties and results in a higher value than the rigid calculation. This relaxation is involved in releasing the strain imposed in the *c*-ZrO_2_ over-layer when it is forced into the registry with the ZrC substrate.

Moreover, Table 3 and Figure 5 show that, using the rigid work of adhesion, the most stable interface model is the *c*-ZrO_2_ (Zr|OO|Zr|OO—) interface model compared to the others. With this interface model, the convergence of the $W_{ad}^{rigid}$ rigid parameter begins at three *c*-ZrO_2_ layers.

#### 3.4.2. Relaxed Work of Adhesion

The relaxed work of adhesion is calculated for all the fully relaxed interfacial systems by allowing the separated ZrC and *c*-ZrO_2_ slabs to fully relax. This parameter characterizes the interfacial bond strengths. In Table 3, the relaxed work of adhesion calculated for the four interface models are summarized. According to Table 3, the interface model involving two layers of oxygen from the ZrO_2_ side (Zr|OO|Zr|OO—) is the most stable in terms of interfacial strength as observed in the calculated relaxed work of adhesion.

Figure 6 provides a pictorial view of the relaxed work of adhesion for all the interface models considered. There is a convergence of the work of adhesion after three layers of ZrO_2_ are deposited on the ZrC. The Wad values initially decrease from one layer to two layers of ZrO_2_ and sharply rise at three layers of ZrO_2_, from which point it converges. Looking at the most stable model (Zr|OO|Zr|OO—), the W_ad_ value sharply increases from 0.251 eV Å^−2^ at two layers of ZrO_2_ to 0.965 eV A^−2^ at three ZrO_2_ layers for the on-top Zr mode of adsorption. This phenomenon is exactly the opposite of what is observed in ceramics deposited on metals where the first one and two layers of deposition are rather stronger than the deposition of three or more layers of ZrO_2_ [22]. Thus, the interfacial strength depends on the first three ZrO_2_ layers deposited. The same feature has been observed for metals deposited on ceramics where the metals are predicted to wet the ceramic surface but then ball up when more than one monolayer of metal is added [48,49,50]. The on-top interface models form stronger interface structures than the fcc models. Weak interfaces are formed when one and two layers of ZrO_2_ are deposited but strong interfaces are obtained when three or more layers of ZrO_2_ are added.

The interface tension defined in Section 2.4.3 is a good parameter for assessing interfacial mechanics and strength. It provides a measure for comparing the strength of bonds at the interface and in the corresponding bulk phases. According to the criteria defined, 0 < ɣ*_int_* < $\sigma_{ZrC}+\sigma_{c-ZrO_{2}}$ is linked to a weakly coupled interface and ɣ*_int_* < 0 to strongly coupled interfaces. With an asymptotic value of $\sigma_{ZrC\left(111\right)}$ = 0.169 eV Å^−2^ and $\sigma_{c-ZrO_{2}}$ = 0.054 eV Å^−2^ combined with *W_ad_* = 0.965 eV Å^−2^ for the most stable interface (Zr|OO|Zr|OO--), the calculated interface tension is ɣ*_int_* = −0.742 eV Å−2. This shows that the interfacial bonds are stronger than the internal bonds in each ceramic bulk phase. When we consider the surface energy for the *c*-ZrO_2_(111) with OO|Zr|OO|Zr|OO termination in an oxygen-rich environment (∆*µ_O_* = 0) as 0.292 eV Å^−2^, the calculated interface tension is −0.504 eV Å^−2^ which is still less than zero. Moreover, ɣ*_int_* ≤ 0 corresponds to a layer-by-layer growth of the ceramic known as the Frank-van-der-Merwe (FM) mode and the mixed-mode is also known as the Stranski-Krastanov (SK) growth mode.

#### 3.4.3. Relaxed Work of Adhesion

A description of the structure and properties of the most stable interface models is provided here. The relaxed stable structures for the Zr|OO|Zr|OO//ZrC(111) interface model using 1, 2, 3, 4, and 5 layers of *c*-ZrO_2_(111) are shown in Figure 7. The interface structure appears to depend somehow on the number of ceramic layers. For this stable interface model, even though the starting geometry was O atoms from the ZrO_2_ side adsorbed directly on top of Zr atoms of the ZrC, the final geometry was O atoms adsorbing at three-fold hollow fcc sites between three Zr (ZrC) atoms. For all layers of ZrO_2_, there is a rearrangement of the Zr|OO|Zr|OO— atoms upon forming the interface into a more stable O|Zr|O|Zr|OO— structure. Thus, the exposed surface of the slab terminates with an O layer.

The crystal shape of the ZrC phase is maintained. However, there is a transformation of the *c*-ZrO_2_ phase into *m*-ZrO_2_. The phase transition is highly evident in the middle of the third, fourth, and fifth layers of the ZrO_2_ interface structures with 3-fold and 4-fold O atoms as well as 6-fold and 7-fold Zr atoms. The (*c*→*m*) transformation is, however, not a complete one due to the restraint imposed on periodic boundary conditions. At T = 0 K, the m-phase is about 7% larger in volume than the *c*-phase and hence this transformation reduces the misfit of about 8% already calculated for this interface model. A similar transformation pattern is found in ceramics deposited on metals [22].

Two types of oxygen bonds are observed at the interface: O1 atoms (ZrO_2_) closer to the interface plane, bonding at fcc sites on the ZrC surface, and O2 atoms (ZrO_2_) above the interface plane, bonding directly on top of Zr (ZrC) atoms. The fcc bonds are exactly the same found in a previous study when the ZrC(111) surface was completely oxidized with a monolayer of oxygen [10]. It was also noticed to passivate the ZrC(111) surface with no further diffusion of oxygen into the bulk. In the 1-layer ZrO_2_ interface, no fcc bonds of the O1 atoms were observed due to the O2 atoms pushing outwards from their bulk positions. These O2 atoms do not show any bonding with the Zr (ZrC) atoms in all layers of ZrO_2_ deposited. The bond distances of the O2 type atoms with the Zr (ZrC) atoms have a minimum of 4 Å. Consequently, at more than three ZrO_2_ layers, there is a structural failure of ZrO_2_ coating on the ZrC substrate. This crack is highly visible in the 4-ZrO_2_ layer interface structure in Figure 7. The crack leaves a monolayer of oxygen deposited on the ZrC. Thus, there is a mixed-mode of deposition of ZrO_2_ on ZrC. A monolayer of oxygen is formed on the ZrC surface and the remaining ZrO_2_ layers ball up. This is not surprising as the very negative interface tension calculated in Section 3.4.2 suggests a mixed-mode of deposition. Additionally, the high work of adhesion, showing an over-adhered interface, results in failure in the ceramic layer, a few angström distances from the interface plane.

Using the 4-layer ZrO_2_ interface model, the calculated bond distances for the O1 atoms at the fcc site of ZrC are d_(O1−r)_ fcc = [2.144–2.169] Å. These distances are in very good agreement with the calculated values of O at fcc sites when the ZrC(111) surface is fully oxidized by a monolayer of oxygen with distances of 2.144 Å and 2.145 Å [10].

### 3.5. Thermodynamic Stability of the Interface

In this section, an analysis of the thermodynamic stability of the different interface models used is considered. The stability of the interface is calculated with respect to the bulk phases and not the surface slabs forming the interface. The interface grand potential $\Omega_{int}^{i/j}$ provides a measure of stability for the different interface models in different terminations. According to Equation (17), the interface grand potential has an interface-dependent term $\emptyset_{int}^{i/j}$ which only differs from the corresponding surface term $\gamma^{i}$ in Equation (4). As such, for the purpose of brevity and clarity, Table 4 provides the $\emptyset_{int}^{i/j}$ values. These terms do not depend on the chemical potentials of excess Zr and O species. In Table 4, it is apparent that the interface is readily formed for all models (due to the negative values of the grand interface dependent terms) with the OO|Zr|OO|Zr--ZrC(111) model being the least stable amongst them. It is, however, evident from Table 4 that the most stable model is the Zr|OO|Zr|OO//ZrC(111) fcc model, contradicting the top Zr|OO|Zr|OO//ZrC(111) model calculated to be the most stable using the relaxed work of adhesion parameters. However, it is worth pointing out that even though the fcc and top models used different starting points of interfacial atom adsorption, the two models resulted in the same configuration with interface O atoms bonding at fcc sites. In addition, the difference between the calculated relaxed work of adhesion values for the two models is only 0.042 eV Å^−2^ using four layers of *c*-ZrO_2_. The interface grand potential thus corroborates the stability criteria established in Section 3.4.2, with Zr|OO|Zr|OO//ZrC(111) being the most stable. When the chemical potentials of excess O and Zr atoms are considered, the calculated $\Omega_{int}^{i/j}$ value, when minimized for every ($∆\mu_{Zr},\;∆\mu_{O}$) pair, resulted in Zr|OO|Zr|OO//ZrC(111) being the most stable model formed.

### 3.6. Interfacial Electronic Properties

#### 3.6.1. Density of States

The electronic features at the interface were analyzed by first considering the density of states when the free surfaces form the interface.

As an initial description of the density of states (DOS) at the interface, a total DOS (TDOS) was obtained by projecting the density of states onto all atoms at the interface and compared with the DOS of the individual separate surfaces. In Figure 8, the TDOS for the most stable interface model, Zr|OO|Zr|OO//ZrC(111) using three layers of ZrO_2_ is shown. Figure 8 also includes the TDOS for the corresponding surface slabs used in constructing this interface model. The valence band maximum is fixed by both ZrC and *c*-ZrO_2_ phases. However, the conduction band minimum is fixed by the ZrC(111) surface upon forming the interface. No new interfacial states are observed in Figure 8 and this is in agreement with the fact that *c*-ZrO_2_ deposits a layer of oxygen on the ZrC(111) surface with the remaining layers breaking off from the interface as explained in Section 3.4.3.

To understand the shift in bands when the surface atoms come together to form the interface, the DOS are projected onto the atoms at the interface and the corresponding atoms in the surface slabs in Figure 9. Upon forming the interface, the Zr (ZrC) conduction bands shift to higher energies as they are filled with electrons from O (ZrO_2_) with a dip at the Fermi level, further stabilizing the interface formed compared to the high level of states at the Fermi level of the corresponding surface. The Zr (ZrC) atoms form a new sharp core state at −18 eV corresponding to the Zr-O bonds formed at the interface. Thus, the Zr (ZrC)—O (ZrO_2_) bond closest to the interface plane is highly localized. There are no significant changes in the C (ZrC) states upon forming the interface with only the valence bands shifting slightly to higher energies. For Zr (ZrO_2_) bands, they are shifted to lower energies upon forming the interface. The sharp Zr (ZrO_2_) band at −15.6 eV which was initially mixed O (ZrO_2_) closest to the interface plane is now lost. This further explains the breakage of the *c*-ZrO_2_ (111) phase after depositing an oxygen layer on the ZrC(111) surface. These Zr (ZrO_2_) bands become diffuse when forming the interface as their bonds with the closest O (ZrO_2_) to the interface plane are broken and the electrons are delocalized. The high surface states of the O (ZrO_2_) closest to the interface plane are quenched and their sharp peaks at −15.6 eV are also lost as they break off from the *c*-ZrO_2_ phase and deposit on the ZrC(111) phase. In the valence band, the O (ZrO_2_) closest to the interface plane interacts strongly with the Zr (ZrC) bands due to their sharp peaks while the O (ZrO_2_) far from the interface plane interacts weakly with its diffuse bands and delocalized electrons.

Figure 10 provides a spatial profile of the electronic structure upon moving from the stable bulk faces to form the exposed surfaces and subsequently the interface. Thus, in Figure 10, the DOS are projected onto the atoms parallel to the interface plane, moving from bulk ZrC to the interface region, then bulk *c*-ZrO_2,_ and finally to the exposed *c*-ZrO_2_(111) surface. The C bands in bulk ZrC remain fairly the same upon forming the interface. In addition, the Zr valence band of ZrC are shifted to lower energies when they form the interface with the introduction of a sharp core state at −18.1 eV which mixes with the O core states of ZrO_2_ closest to the interface plane. At the interface region, the valence band is a mixture of Zr (ZrC) d-bands and O (ZrO_2_) p-bands with a covalent bonding nature. There is a clearly diffuse interaction of the O (ZrO_2_) p-bands far from the interface plane, further explaining the weak interaction of the second O layer (ZrO_2_) far from the interface plane when the interface is formed. It is also apparent that the O (ZrO_2_) p-bands far from the interface maintain the bulk nature of such bands with a similar feature for the Zr (ZrO_2_) d-bands far from the interface plane. The conduction band in this bulk ZrO_2_ region is mainly Zr-d states. Moving to the exposed surface on the ZrO_2_ side, the Zr-d states in the conduction band are reduced, shifted to lower energies, and new states appear close to the Fermi level while the valence band becomes less diffuse. The exposed ZrO_2_ surface is stabilized by the oxygen termination as there are relatively no states around the Fermi level. This termination is the same foundation for the most stable exposed surface of *c*-ZrO_2_(111).

#### 3.6.2. Charge Transfer Analysis

A Bader charge analysis is provided in this section to understand how charges are transferred when forming the interface from the cleaved surfaces. The charge transfer on the atoms is arranged in a spatial profile of the atoms from the surface to respective layers in the different phases forming the interface as described by Christensen and Carter [21]. As a first step, we provide a charge transfer analysis for the most stable interface with different layers of *c*-ZrO_2_(111) starting from one to five in a spatial profile and finally analyzing the four *c*-ZrO_2_(111) layer stable interface model, Zr|OO|Zr|OO//ZrC(111) top site. Table 5 shows that upon forming the interface from the free surfaces, there is a significant amount of charge transfer in the interfacial region.

Nearly all the charge movements are from the cations (Zr) on the ZrC side of the interface plane to the O atoms (ZrO_2_) closest to the interface plane. Initially, with one layer of ZrO_2_ deposited, there is the transfer of approximately 0.50 e^−^ from Zr (ZrC) to O atoms on the ZrO_2_ side of the interface. This value increases with the number of layers and becomes constant at three layers of ZrO_2_ deposited. The high charge transfer of more than 0.90 e^−^ from Zr (ZrC) to O (ZrO_2_) closest to the interface results in very strong interfacial bonds between the interface Zr and O atoms, subsequently causing breakage of the O layer (ZrO_2_) closest to the interface from the remaining part of the ZrO_2_ phase. In all the layers of ZrO_2_ deposited, there is essentially no significant charge redistribution in the atomic layers farther away from the interface moving toward the bulk region of the respective phases. With three or more ZrO_2_ layers deposited, the high charge transferred from interface cations to anions releases part of the axial strain imposed on the ZrO_2_ phase as it is brought into registry with the ZrC substrate upon forming the interface. The magnitude of the charges transferred from Zr (ZrC) to O (ZrO_2_) for the different layers of ZrO_2_ is in the same pattern as the relaxed work of adhesion computed for the different numbers of layers. In Table 6, a spatial profile of the charge transfer is provided to obtain a further understanding of the electronic structure of the interface formed and how it affects the corresponding bulk and surface properties. The most stable Zr|OO|Zr|OO—ZrC(111) top site model with four layers of ZrO_2_ is used in this analysis. In this table, the average charge distribution per layer is arranged in a profile moving from the interface plane towards the bulk and then the exposed surface regions. In this analysis, it is essential to compare the ∆*Q_s_* values of Table 6 when analyzing the surface regions (with respect to atoms in the corresponding surface slabs), while ∆*Q_b_* values are used when making bulk comparisons (with respect to corresponding atoms in the bulk phases). In the interfacial region, there is a significant amount of charge transfer, compared to the corresponding free surface. This high amount of charge distribution mainly originates from Zr (ZrC) and O atoms (ZrO_2_) at the interface. However, moving away from the interface into the bulk regions of both phases, and finally, to the exposed surface area of ZrO_2_, there is virtually no charge redistribution. The ∆*Q_b_* values for the bulk regions also show no charge redistribution when forming the interface. This further ascertains the interface to be affected only by atomic layers closer to the interface plane.

## 4. Summary and Conclusions

Different theoretical approaches have been used to characterize the oxide layer formed on ZrC nanoparticles. Finite temperature ab initio molecular dynamic was used to build two layers of ZrO_2_ on a ZrC(111) surface. An ordered layer of ZrO_2_ was observed to form on the ZrC(111) surface. Then, periodic DFT was used to characterize the interface between ZrC(111) and *c*-ZrO_2_(111). The preferred interface observed was formed between Zr terminated ZrC(111) and an OO-terminated *c*-ZrO_2_(111) leading to a final interface model Zr|OO|Zr|OO//ZrC(111). The main mechanical property used to characterize the interface was the relaxed work of adhesion *W_ad_*. This value reveals the Zr|OO|Zr|OO//ZrC(111) model as the most stable interface with a high *W_ad_* value compared to the other models. A thermodynamic analysis investigating the interface grand potential also confirmed the Zr|OO|Zr|OO//ZrC(111) model as the most stable interface. A close examination of the structural properties revealed the deposition of an oxygen layer from the ZrO_2_ phase onto the ZrC(111) surface with the remaining part of ZrO_2_ breaking off from the interface, suggesting a crack at this interface. This also was in very good agreement with a previous study on the oxidation of ZrC(111) surfaces which revealed the formation of an oxygen layer on the surface with the further addition of O atoms leading to an endothermic reaction. In addition, the electronic structure of the interface was analyzed using the density of states calculated for the interface models. The DOS showed no major changes induced by the formed interface but only features of the deposited oxygen layer. In using the Bader charge analysis, an enormous amount of charge transfer at the interface was discovered, originating mainly from cations on the ZrC side (Zr) to the O atoms from ZrO_2_ at the interfacial region. Moreover, there was virtually no charge redistribution in the bulk regions of the interface slab. Thus, the interfacial properties were governed by local effects, only confined to the first two atomic layers around the interface plane. To conclude, this study brought new insights into the structure and composition of the surface/interface of ZrC. This will pave the way for new opportunities in the field of hybrid materials with high-performance properties in harsh environments.

## Figures and Tables

**Figure 1 molecules-27-02954-f001:**
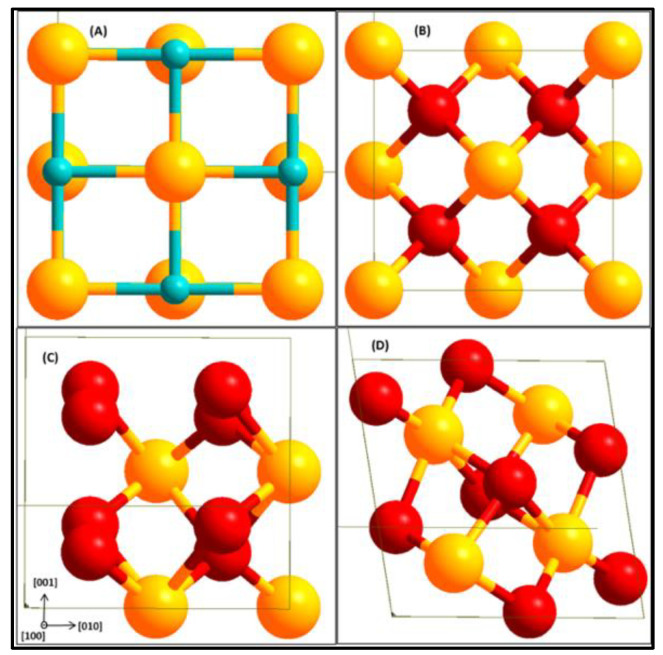
Bulk structures of ZrC (**A**), *c*-ZrO_2_ (**B**), *t*-ZrO_2_ (**C**), and *m*-ZrO_2_ (**D**). Yellow (Zr), light blue (C), and red (O).

**Figure 2 molecules-27-02954-f002:**
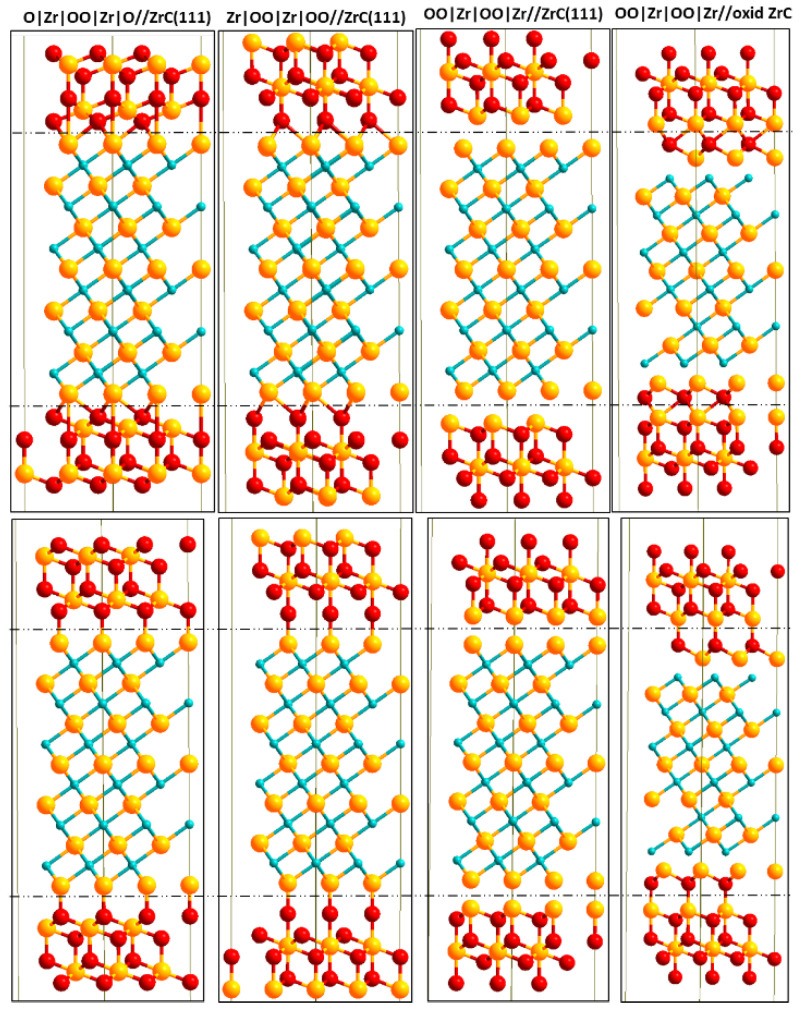
Side views of different interface model slabs using two layers of *c*-ZrO_2_ (111). Top structures are models with fcc bonding sites at the interface with corresponding on-top interface bonding sites in the bottom structures. Yellow (Zr), light blue (C), and red (O).

**Figure 3 molecules-27-02954-f003:**
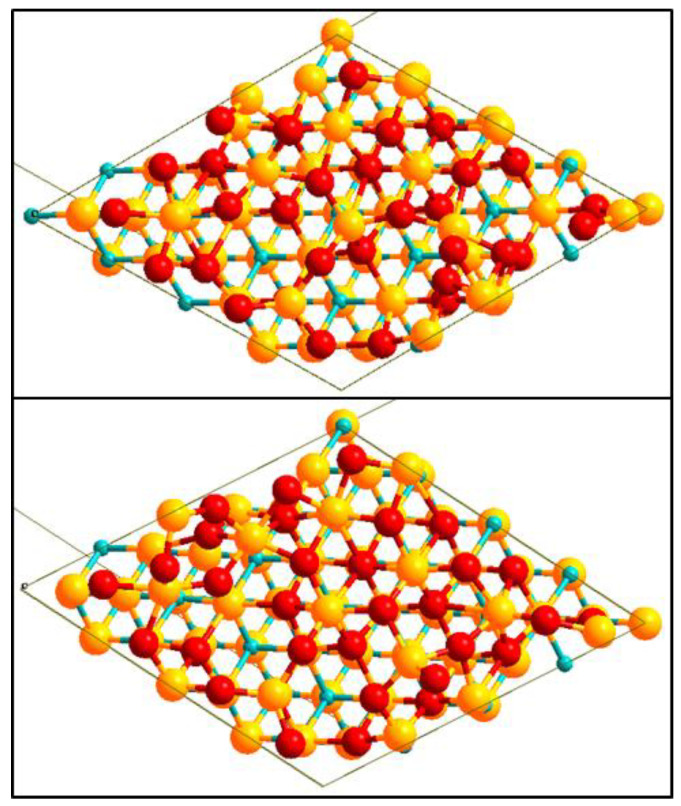
The high-temperature structure after 20 ps (top) and low temperature, T = 500 K (bottom) from MD simulation. The global effect of the temperature reduction is an increase in the symmetry of the ZrO_2_ layer. Yellow (Zr), light blue (C), and red (O).

**Figure 4 molecules-27-02954-f004:**
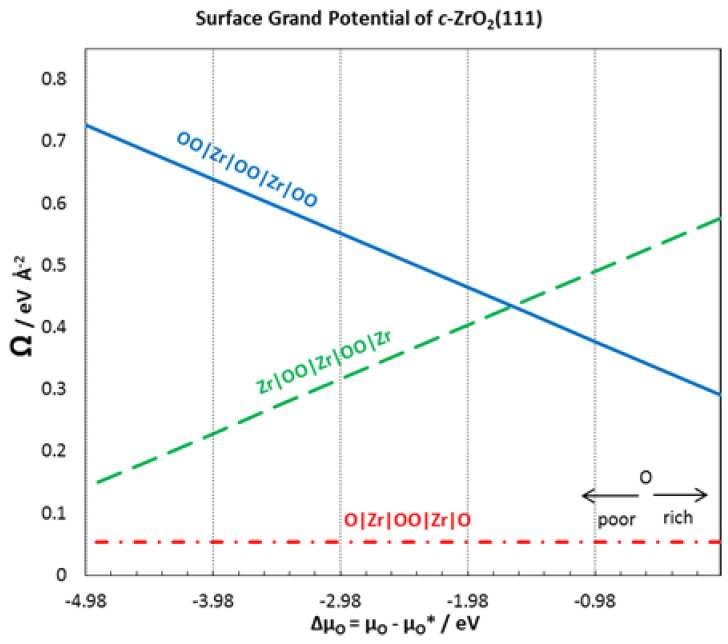
Surface grand potential for different terminations of *c*-ZrO_2_(111) surface.

**Figure 5 molecules-27-02954-f005:**
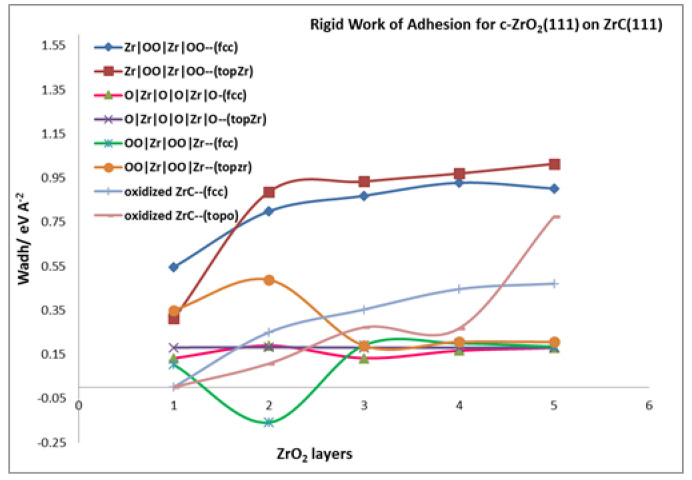
Rigid Work of adhesion for different interface models of *c*-ZrO_2_(111) on ZrC(111) surface.

**Figure 6 molecules-27-02954-f006:**
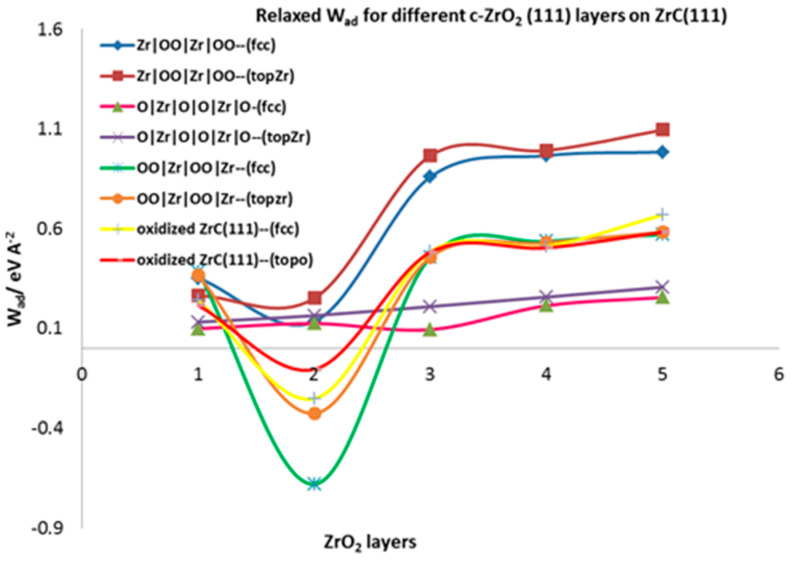
Relaxed Work of adhesion for different interface models of *c*-ZrO_2_ (111) on ZrC(111) surface.

**Figure 7 molecules-27-02954-f007:**
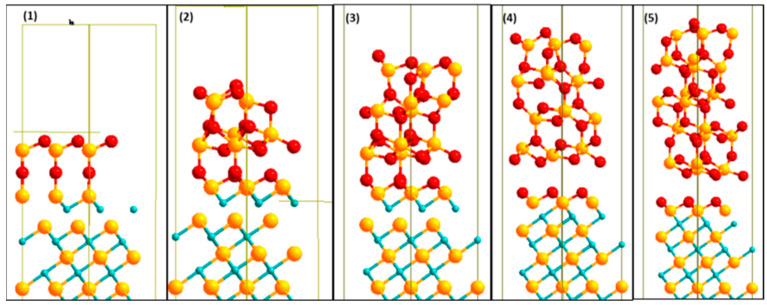
Relaxed stable structures for 1, 2, 3, 4, and 5 (from **left** to **right**) *c*-ZrO_2_(111) deposited on Zrc(111). The most stable interface model Zr|OO|Zr|OO—ZrC(111) is shown here. Yellow (Zr), light blue (C), and red (O).

**Figure 8 molecules-27-02954-f008:**
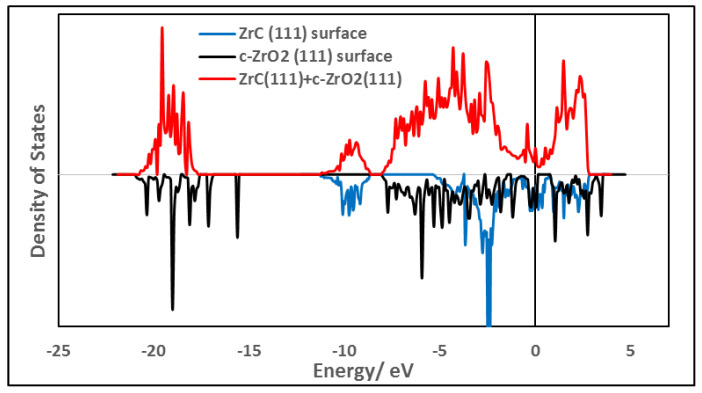
Comparison of the TDOS of interfacial structure Zr|OO|Zr|OO—ZrC(111) (on **top**) with the TDOS of the bare surfaces (on the **bottom**). An interface model with four layers of *c*-ZrO_2_(111) is used.

**Figure 9 molecules-27-02954-f009:**
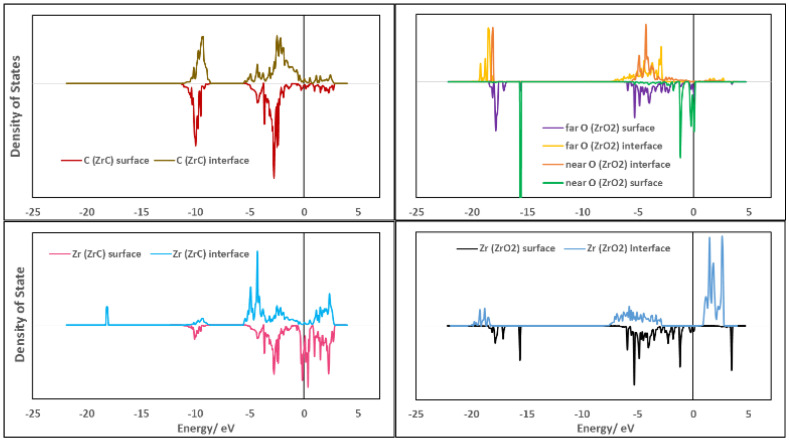
Comparison of the PDOS of each atom at the interface and the corresponding surface slab for the Zr|OO|Zr|OO—ZrC(111) top site interface model. The upper part of the spectra of each plot are for atoms in the interface structure and those at the lower part are for the atoms in the corresponding surface slabs. Atoms labeled as near are closer to the interface plane than atoms labeled as far.

**Figure 10 molecules-27-02954-f010:**
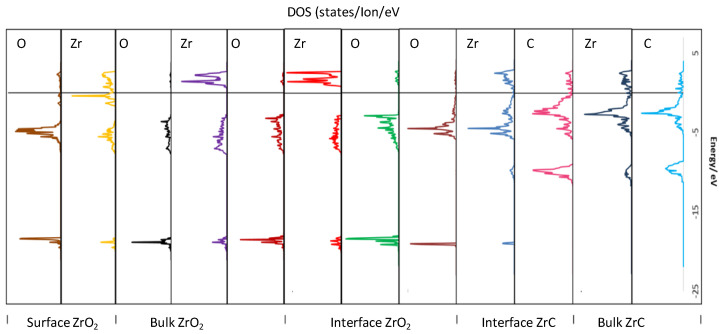
PDOS of atoms at the interface, in the bulk, and the exposed surface of the four *c*-ZrO_2_ layer interface slab (Zr|OO|Zr|OO—ZrC(111)). The states are aligned parallel to the interface plane with each region marked on the top layer. The Fermi level is aligned at the zero energy position.

**Table 1 molecules-27-02954-t001:** Surface mismatch parameter *ϒ* calculated for different combinations of ZrC and *c*-ZrO_2_ surface.

ZrC	*c*-ZrO_2_	Overlap Area(S_1–2_)/Å^2^	Misfit (ϒ)
(111)	(001)	44.874	0.169
(111)	(110)	44.874	0.157
(111)	(111)	44.874	0.072

**Table 2 molecules-27-02954-t002:** Calculated surface energies for O|Zr|OO|Zr|O termination of *c*-ZrO_2_(111) surface at all O chemical potential.

Number of Layers	1	2	3	4	5	6
*Ω^i^*/eV Å^−2^	0.068	0.053	0.055	0.055	0.054	0.054

**Table 3 molecules-27-02954-t003:** Rigid, $W_{ad}^{rigid}$ and relaxed, $W_{ad}^{relaxed}$ work of adhesion for different interface models using different numbers of *c*-ZrO_2_ (111) layers at both fcc and on-top adhesion sites at the interface region.

Interface Model	OO|Zr|OO|Zr—ZrC(111)		Zr|OO|Zr|OO—ZrC(111)		O|Zr|OO|Zr|O—ZrC(111)		OO|Zr|OO|Zr—4O-ZrC(111)	
*c*-ZrO_2_(111) Layers	fcc	top	fcc	top	fcc	top	fcc	top
Rigid work of Adhesion, $W_{ad}^{rigid}$								
1	0.104	0.350	0.545	0.312	0.132	0.182	0.002	0.002
2	−0.158	0.488	0.798	0.886	0.190	0.182	0.250	0.108
3	0.191	0.190	0.869	0.934	0.133	0.181	0.353	0.273
4	0.200	0.207	0.928	0.970	0.67	0.180	0.446	0.268
5	0.185	0.206	0.902	1.014	0.180	0.181	0.471	0.776
Relaxed work of Adhesion, $W_{ad}^{relaxed}$								
1	0.383	0.366	0.351	0.264	0.098	0.131	0.240	0.215
2	−0.682	−0.327	0.134	0.251	0.124	0.164	−0.252	−0.106
3	0.458	0.457	0.859	0.965	0.093	0.209	0.484	0.481
4	0.536	0.529	0.966	0.991	0.214	0.257	0.511	0.503
5	0.570	0.582	0.983	1.095	0.254	0.306	0.669	0.583

**Table 4 molecules-27-02954-t004:** Interface dependent terms of the interface grand potential $\emptyset_{int}^{i/j}$.

Model Site	$\emptyset_{int}^{i/j}$/eV Å^−2^
O|Zr|OO|Zr|O-ZrC(111)	
fcc	−0.562
top	−0.605
Zr|OO|Zr|OO-ZrC(111)	
fcc	−1.555
top	−0.451
OO|Zr|OO|Zr-ZrC(111)	
fcc	−0.257
top	−0.251
Zr|OO|Zr-oxidized ZrC(111)	
fcc	−0.444

**Table 5 molecules-27-02954-t005:** Charge transfer analysis of the interfacial structure between ZrC(111) surface and different numbers of layers of *c*-ZrO_2_(111) in the Zr|OO|Zr|OO—ZrC(111) top site interface model. Values reported are net charges (electrons/atom) obtained with respect to the charges on the atom in the corresponding surface slabs that form the interface.

	Ion Type		ZrO_2_ Layers on ZrC				
			1	2	3	4	5
ZrO_2_ layer at Interface		Zr	0.05	0.01	−0.09	−0.07	−0.08
	O		0.04	0.42	0.21	0.25	0.23
		O	0.38	0.73	0.67	0.75	0.75
Interface Plane							
ZrC layer at Interface	Zr		−0.49	−0.89	−0.94	−0.92	−0.93
		C	0.00	−0.04	−0.05	−0.07	−0.07

**Table 6 molecules-27-02954-t006:** The spatial profile of charge transfer is arranged along the normal direction of the interface. ∆*Q_s_* is the difference in the charge of the ion with the corresponding ion in the isolated surface slab used to create the interface, and ∆*Q_b_* is the difference in charge between the ion and the corresponding ion in the bulk structure.

Layer in Slab	4 ZrO_2_ (Surface)			3 ZrO_2_ (Bulk)			2 ZrO_2_ (Bulk)			1 ZrO_2_ (Interface)			1 ZrC (Interface)		2 ZrC (Bulk)	
Ion Type	O	Zr	O	Zr	O	O	Zr	O	O	Zr	O	O	Zr	C	Zr	C
Absolute *Q*	7.14	2.63	7.18	1.79	7.17	7.11	1.72	7.09	7.11	1.72	7.13	7.17	2.00	5.74	2.28	5.71
∆*Q_s_* (vs. surface)	−0.04	0.07	0.01	0.04	0.09	−0.08	−0.01	−0.02	−0.07	−0.07	0.25	0.75	−0.92	−0.07	−0.07	−0.03
∆*Q_b_* (vs. bulk)	−0.01	0.94	0.03	0.10	0.02	−0.04	0.03	−0.06	−0.04	0.03	−0.02	0.02	−0.34	0.08	−0.06	0.05

## Data Availability

Data is contained within the article or the Appendix A.

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
