# Peer review of "Characterizing the ZrC(111)/c-ZrO2(111) Hetero-Ceramic Interface: First Principles DFT and Atomistic Thermodynamic Modeling"

_molecules, 2022, doi:10.3390/molecules27092954_

Round 1
Reviewer 1 Report
Addad and co-workers present a computational study on the characterization
of the ZrC(111)/c-ZrO2(111) hetero-ceramic interface at the PBE/PAW level of theory.
Overall, the manuscript is well-written and easy to follow. Even though I am not an
expert on this specific solid-state chemistry there are some points that may have
to be addressed before consideration for publication.
1.) The authors name the PBE functional explicitly but do not mention any
London dispersion treatment. From my experience London dispersion (e.g. modeled by Grimme's D3 or D4 model)
are crucial to correctly describe solids, and specifically interface interaction energies correctly.
If no dispersion correction was used, the calculations should be redone accordingly using e.g. DFT-D4
2.) The theoretical level used in the MD simulations is unclear and should be specified in the corresponding section.
3.) How do the authors ensure that the size of their model systems is converged? This may be explained in more detail and supplementary
data should be provided.
4.) The authors present a Bader charge transfer analysis. Which density is used here, if the GGA density was used, how prone is it to
over delocalization effects?
5.) Figure 1. Labelling the atoms (or including a key) directly would enhance readability (without having to check the caption first).
6.) Figure 6. Is it somehow possible to highlight the atoms, that moved significantly? The differences between both subfigures are hard to
figure out at first glance. The overall simulation time of the MD should be added to the caption.
Overall, the manuscript may become suitable for publication after major revision. Nevertheless, another experts opinion on the experimental
background may be needed here.
Author Response
We would like to thank you for your letter and the very interesting comments concerning our manuscript entitled “Characterizing the ZrC(111)/c-ZrO2(111) Hetero-ceramic In-terface: First Principles DFT and Atomistic Thermodynamic Modeling”. These comments were valuable and very helpful to improve our research. The answers are indicated below.
------
1.) The authors name the PBE functional explicitly but do not mention any
London dispersion treatment. From my experience London dispersion (e.g. modeled by Grimme's D3 or D4 model)
are crucial to correctly describe solids, and specifically interface interaction energies correctly.
If no dispersion correction was used, the calculations should be redone accordingly using e.g. DFT-D4
Thank you for your remark. As it is noticed, the DFT calculations performed in the study do not take into account explicitly the London correction. However, the effect of this correction will be very small in this study and will not modify the conclusion. Indeed, the interaction between the surfaces are covalent ones which are well described by DFT. The addition of Grimme’s correction slightly increases the surface energies but will have almost no effect on the interaction. The number of atoms in the region used to compute the D3 corrections will be almost the same in the buck and at the interface. There will be an almost perfect compensation.
2.) The theoretical level used in the MD simulations is unclear and should be specified in the corresponding section.
Thank you for your comment. The AIMD calculations have been performed mostly to generate quite rapidly interface configurations. The calculation has been performed in the NVE ensemble with a time steps of 1 fs. The temperature has been increased progressively up to 1000 K by re-scaling the kinetic energy every 20 steps. Then the calculation have been done for 20 ps. The indication has been added in the text.
3.) How do the authors ensure that the size of their model systems is converged? This may be explained in more detail and supplementary
data should be provided.
Thank you for your question. The effect of the parameter on the calculation have been tested in the previous papers. The DOS of ZrC in the middle of the ZrC layer is similar to the bulk one. It is a good indication that the slab is thick enough to reproduce the buck constrains.
4.) The authors present a Bader charge transfer analysis. Which density is used here, if the GGA density was used, how prone is it to
over delocalization effects?
Thank you for your question. The Bader charges have been computed using the GGA density. The GGA calculation may indeed overestimate the delocalisation. However, in the study, we focus on the evolution of the charges and not on the value of the charge. In this case, the effect of the overestimation of the delocalisation should be small.
5.) Figure 1. Labelling the atoms (or including a key) directly would enhance readability (without having to check the caption first).
Thank you for your remark. The colors for O and C are classical ones. We do not think that it is helpful to add the name on the figure. The structure would be less visible.
6.) Figure 6. Is it somehow possible to highlight the atoms, that moved significantly? The differences between both subfigures are hard to
figure out at first glance. The overall simulation time of the MD should be added to the caption.
Thank you for your comment. The length of the simulation was added in the caption. Almost all the top atoms move between the 2 structures even if the displacement is small. It is difficult to enhance only few displacements. However, it can be seen that the low temperature structure is more symmetric than the high one. A comment was added in the caption.
Reviewer 2 Report
Osei-Agyemang et al. report the detailed study on the formation of ZrC/ZrO2 interface with different methods, ie. geometry optimization, electronic structures analysis, MD simulations. The work provides some useful information on the coating structure and properties of ZrO2 on ZrC. However, it is poorly written with a lot of grammar mistakes. the following issues need to be addressed:
1 for a clear understanding, please provide what type MD simulations (NPT or NVT) were performed? how many layers were fixed. The dimensions of the box and the number of atom
How long the MD have been carried out. what bulk calculations were performed?
2 From Figure7, it looks like the ZrO2 left the surface instead of staying there. Please provide bonding, charge density analysis on the interface to show whether it is stable.
3 The DOS figure with upside down curves is difficult to read and confusion.
3 please proofread the whole article: example mistakes:
“was studied by finite temperature molecular dynamics simulation and DFT “ “a serious problem encountered with this material, when used in harsh conditions, is the proneness to oxidation. “
“As the composition of the surface will condition the further functionalization that could involve preceramic precursors such as polycarbosilanes, it is necessary to study the oxidized layer, provide detail analysis on the structure, energetics, and stability at the interfacial region between ZrC and the oxidized lay “
“section 2 provides details on the finite temperature molecular dynamics simulation and general calculation parameters as well as procedures for building the interface are provided “
“Electron density functional calculations (DFT) “

Author Response
We would like to thank you for your letter and the very interesting comments concerning our manuscript entitled “Characterizing the ZrC(111)/c-ZrO2(111) Hetero-ceramic In-terface: First Principles DFT and Atomistic Thermodynamic Modeling”. These comments were valuable and very helpful to improve our research. The answers are indicated below.
-------
1 for a clear understanding, please provide what type MD simulations (NPT or NVT) were performed? how many layers were fixed. The dimensions of the box and the number of atom
Thank you for your question. The AIMD was performed in the NVE ensemble. The objective of the calculation was to generate structures and not to compute thermodynamic properties. The total number of atoms included in the simulation is 60 (i.e. 28 Zr, 16 C and 16 O). The size of the cell was added in the text.
How long the MD have been carried out. what bulk calculations were performed?
The production run is 20 ps.
2 From Figure7, it looks like the ZrO2 left the surface instead of staying there. Please provide bonding, charge density analysis on the interface to show whether it is stable.
Thank you for your comment. The lack of bonds in the drawing is due to the drawing program parameters. The value of the interface energy is given in table 3.
3 The DOS figure with upside down curves is difficult to read and confusion.
Thank you for your remark. The figure was changed.
3 please proofread the whole article: example mistakes:
“was studied by finite temperature molecular dynamics simulation and DFT “ “a serious problem encountered with this material, when used in harsh conditions, is the proneness to oxidation. “
“As the composition of the surface will condition the further functionalization that could involve preceramic precursors such as polycarbosilanes, it is necessary to study the oxidized layer, provide detail analysis on the structure, energetics, and stability at the interfacial region between ZrC and the oxidized lay “
“section 2 provides details on the finite temperature molecular dynamics simulation and general calculation parameters as well as procedures for building the interface are provided “
“Electron density functional calculations (DFT) “
Thank you for your remark. The manuscript was proofread, and the mentioned sentences were changed.
Reviewer 3 Report
In word

Author Response
We would like to thank you for your letter and the very interesting comments concerning our manuscript entitled “Characterizing the ZrC(111)/c-ZrO2(111) Hetero-ceramic In-terface: First Principles DFT and Atomistic Thermodynamic Modeling”. These comments were valuable and very helpful to improve our research. The answers are indicated below.
-------
This paper studied the aCharacterizing the ZrC(111)/c-ZrO2(111) Hetero-ceramic Interface
based on first principle DFT and and Atomistic Thermodynamic Modeling. It provided an
effect way to brought the new insights in the structure and composition of the
surface/interface of ZrC.
1. Please analyze in detail why the surface energies of c-ZrO2(111) surface with
O|Zr|OO|Zr|Otermination converges after 3 layers in “3.3 Surface and bulk properties
of ZrC and ZrO2”.
The relaxation of a surface involved in general 3 layers. The first interlayer distance was reduced due to the lack of bond at the surface. The 2nd interlayer distance increased, and the 3rd interlayer distances were almost unchanged. The following reference is a good example on iron oxide: Surface Science, 302 (1994) 259-279.
- Please analyze in detail why the work of adhesion convergenced after three layers of
ZrO2 deposited on the ZrC.
The reason is similar to the previous one.
- The table for this article is not standard, please redraw it
Thank you for your remark. The tables are now standard.
Round 2
Reviewer 1 Report
The authors adressed most of the points. Nevertheless, I think the authors should comment on the London dispersion point in the manuscript. Further, I disagree about the labels in the figures. In most of the molecular chemistry community, carbon is depicted in grey not in light blue. Therefore, labelling only one atom or adding a key (really not much work!?) would enhance the figure. The paper may be accepted after minor revision.
Author Response
We would like to thank you for your letter and the very interesting comments concerning our manuscript entitled “Characterizing the ZrC(111)/c-ZrO2(111) Hetero-ceramic Interface: First Principles DFT and Atomistic Thermodynamic Modeling”. These comments were valuable and very helpful to improve our research. The answers are indicated below.
------
The authors adressed most of the points.
Nevertheless, I think the authors should comment on the London dispersion point in the manuscript.
Thank you for your remark. Two sentences were added in the structural models and calculation section.
Further, I disagree about the labels in the figures. In most of the molecular chemistry community, carbon is depicted in grey not in light blue. Therefore, labelling only one atom or adding a key (really not much work!?) would enhance the figure.
Thank you for your comment. The labels are already described in the figure caption. The key was also added to Figures 2, 3 and 7.
Reviewer 2 Report
The systems is not magnetic. It is suggested to plot the DOS on the same side of the axis.
The author needs to carefully proofread the article. there are still several grammar mistake, in particularly, the introduction section.
Author Response
We would like to thank you for your letter and the very interesting comments concerning our manuscript entitled “Characterizing the ZrC(111)/c-ZrO2(111) Hetero-ceramic Interface: First Principles DFT and Atomistic Thermodynamic Modeling”. These comments were valuable and very helpful to improve our research. The answers are indicated below.
-------
The systems is not magnetic. It is suggested to plot the DOS on the same side of the axis.
Thank you for your comment. In figure 8, we compare the DOS of the reference surfaces with the DOS of the interface. The comparison will be difficult if we put them on the same graph. The caption has been modifed to avoid misunderstanding with magnetic systems.
-------
The author needs to carefully proofread the article. there are still several grammar mistake, in particularly, the introduction section.
Thank for your remark. The manuscript has been proofread by an English native translator/interpreter.